# Preparation and Vasodilation Mechanism of Angiotensin-I-Converting Enzyme Inhibitory Peptide from *Ulva prolifera* Protein

**DOI:** 10.3390/md22090398

**Published:** 2024-08-31

**Authors:** Zhiyong Li, Hongyan He, Jiasi Liu, Huiyue Gu, Caiwei Fu, Aurang Zeb, Tuanjie Che, Songdong Shen

**Affiliations:** 1School of Biology & Basic Medical Sciences, Soochow University, Suzhou 215101, China; 2Suzhou Chien-Shiung Institute of Technology, Suzhou 215101, China; 3Key Laboratory of Functional Genomic and Molecular Diagnosis of Gansu Province, Lanzhou 730030, China

**Keywords:** *Ulva prolifera* protein, ACE inhibitory peptide, Akt, vasodilation mechanism

## Abstract

*Ulva prolifera*, a type of green algae that can be consumed, was utilized in the production of an angiotensin-I converting enzyme (ACE) inhibitory peptide. The protein from the algae was isolated and subsequently hydrolyzed using a neutral protease. The resulting hydrolysate underwent several processes including Sephadex-G100 filtration chromatography, ultrafiltration, HPLC-Q-TOF-MS analysis, ADMET screening, UV spectrum detection test, molecular docking, and molecular dynamic simulation. Then, the ACE inhibitory peptide named KAF (IC_50_, 0.63 ± 0.26 µM) was identified. The effectiveness of this peptide in inhibiting ACE can be primarily attributed to two conventional hydrogen bonds. Additionally, it could activate endothelial nitric oxide synthase (eNOS) activity to promote the generation of nitric oxide (NO). Additionally, KAF primarily increased the intracellular calcium (Ca^2+^) level by acting on L-type Ca^2+^ channel (LTCC) and the ryanodine receptor (RyR) in the endoplasmic reticulum, and completed the activation of eNOS under the mediation of protein kinase B (Akt) signaling pathway. Our study has confirmed that KAF has the potential to be processed into pharmaceutical candidate functions on vasoconstriction.

## 1. Introduction

Hypertension has emerged as a global health concern, affecting more than 1.13 billion people worldwide, according to a survey conducted by the World Health Organization [1]. The renin–angiotensin–aldosterone regulatory system (RAAS) plays a crucial role in the regulation of blood pressure [2]. Within this system, ACE is one of the key enzymes, catalyzes the conversion of inactive Angiotensin I (Ang I) into the potent vasoconstrictor Angiotensin II (Ang II), thereby deactivating the vasodilator bradykinin [3]; this highlights the effectiveness of ACE inhibitors as an important therapeutic approach for the treatment of hypertension. Small molecules can enter the active pocket and inhibit the activity of ACE [4]. Currently, bioactive peptides with ACE inhibitory activity and no side effects have garnered significant attention. Various animal and plant proteins, such as *Mytilus edulis* [5], sea cucumber (*Apostichopus japonicus*) gonads [6], skipjack tuna (*Katsuwonus pelamis*) muscle [7], and *Takifugu flavidus* [8] have been reported to contain ACE inhibitory peptides. These peptides hold promise as natural alternatives for managing hypertension.

*Ulva prolifera*, an edible green alga, thrives in the coastal areas of China. However, over the past decade, excessive nutrients in the water have caused recurring outbreaks of green tides dominated by *U. prolifera*. These tides lead to the accumulation of algae on the seabed, resulting in water hypoxia and the subsequent death of aquatic organisms [9]. Cleaning up excessive *U. prolifera* requires significant manpower and resources. Therefore, it is essential to explore ways to transform this alga into a valuable resource. A previous study indicated that *Enteromorpha spp*. (one of *U. prolifera*) boasts a substantial biomass and possesses an excellent nutritional composition, including proteins (9–14%), ether extract (2–3.6%), ash (32–36%), and a total fatty acid content of 10.9 g/100 g [10]. These characteristics make *U. prolifera* a promising candidate for various commercial applications. Previous studies have shown that the enzymatic hydrolysis of *U. prolifera* proteins can generate small-molecule peptides with ACE inhibitory activity [11]. However, the mechanism behind the vasodilation effects of *U. prolifera* peptides has not been reported before.

The endothelium, a cellular monolayer lining the blood vessel wall, plays a crucial role in maintaining overall health and homeostasis in multiple organs [12]. Endothelial cells are involved in regulating blood pressure through the secretion of various cytokines. However, metabolic disorders can lead to endothelial dysfunction and damage in the body. Therefore, in vitro cultivation of vascular endothelial cells is an important method of studying the mechanism of ACE inhibitory peptides [13,14]. Nitric oxide (NO), produced by various organs and tissues, has a blood-pressure-lowering effect and enhances male sexual function. Previous studies have reported that small molecules can activate the phosphorylation of eNOS in endothelial cells, thereby promoting NO synthesis [15]. Therefore, it is important to explore the function of eNOS activation and NO production by *U. prolifera* peptides.

The objective of this study was to develop a promising ACE inhibitory peptide from *U. prolifera*. Additionally, the inhibitory pattern and molecular interaction mechanism of the peptide were investigated. Furthermore, the cellular mechanism by which the purified peptide regulates blood pressure was explored using human umbilical vein endothelial cells (HUVECs).

## 2. Results and Discussion

### 2.1. Purification and Identification of ACE Inhibitory Peptides from U. prolifera Protein Hydrolysate

Some proteins can produce ACE inhibitory peptides in the digestion of neutral protease, such as Pacific saury [16], tuna processing by-products [17], and Bovine casein [18]. It has also been reported that *U. prolifera* protein contains peptides with ACE inhibitory activity [11]. Therefore, in this study, *U. prolifera* protein was hydrolyzed using neutral protease under optimal conditions. 

Then, a hydrolysate with ACE inhibitory activity was obtained, with degrees of hydrolysisi (DH) and IC_50_ of 33.59% and 1471.92 μg/mL, respectively (Figure 1). DH is a parameter that describes the degree of *U. prolifera* protein hydrolysis, which is higher than in the previous report [11]. The result indicates that a large amounts of peptides and amino acids were released from *U. prolifera* protein.

To obtain a more efficient fraction, the hydrolysate was separated using the Sephadex-G100. The separation range of Sephadex-G100 is 4000–15,000 Da, and it can exclude undecomposed proteins. Additionally, peptides larger than the gel mesh of Sephadex-G100 flow out of the gap quickly. But, instead, peptides smaller than the gel mesh can be repeatedly shuttled through each gel particle before eventually exiting the column. Fraction 2 has a longer separation time and therefore contains more small peptides. Subsequently, Fraction 2 with the highest ACE inhibitory activity was further purified by an ultrafiltration membrane (Figure 1). Among the three fractions, the one with a molecular weight less than 3 kDa showed the highest ACE inhibitory activity, with an IC_50_ value of 26.36 μg/mL (Table 1). These findings support the notion that smaller peptides are more likely to enter the active center and possess stronger ACE inhibitory activity [11]. To determine the amino acid sequences of this fraction (<3 kDa), HPLC-MS/MS analysis was performed, resulting in the identification of 2257 peptides by comparing the data with those available in the NCBI database (Appendix A).

### 2.2. Screening of the Potential ACE Inhibitory Peptides

ADMET has been widely used to screen the highest ACE inhibitory peptides [4]. As shown in Table 1, thirteen peptides exhibited desirable ADMET parameters, including non-toxicity, good water solubility, high biological activity, high human intestinal absorptivity, and high blood–brain barrier permeability. However, among these peptides, only KAF can dock with ACE according to molecular docking. The MS/MS spectra of this peptide is shown in Figure 2. In vitro ACE inhibitory assay revealed that KAF exhibited an IC_50_ value of 0.63 ± 0.26 µM. Interestingly, KAF had a higher -C Dock Energy than captopril (0.025 µM) but less ACE inhibitory activity [19]. This discrepancy may be attributed to the fact that captopril, with its smaller molecular size, is more likely to interact with the ACE enzyme.

### 2.3. Inhibition Pattern of Peptide against ACE

To investigate the inhibition mechanism of KAF, it was co-incubated with the substrate (HHL) and ACE under the optimum conditions. Lineweaver–Burk plots suggested KAF was a competitive inhibitor (Figure 3). Similarly, other peptides from natural product protein, such as EACF [4], QDVL [15], and AEYLCEAC [20], also exhibited competitive inhibition. These peptides interacted with the active site of ACE, preventing the generation of products.

### 2.4. Molecular Docking and Dynamics Simulation

Molecular docking was performed to explore the interaction mechanism between candidates and ACE. As shown in Figure 4, the stabilized poses of KAF are attributable to five types of bonds: attractive charge, conventional hydrogen, carbon hydrogen, pi-alkyl, and metal-acceptor bonds. Among these, the three conditional hydrogen bonds and the key binding sites Ala354 and Asp415 played an important role in stabilizing KAF-ACE complex (Figure 4). Additionally, KAF was able to bind the active site Zn701 of ACE by attractive charge and metal-acceptor bonds, which aligns with the kinetics result indicating that KAF, as a competitive inhibitor, can enter the active pocket of ACE. In comparison, the control group Captopril formed one hydrogen bond with ACE and Captopril (Figure 4). However, Captopril (IC_50_, 0.025 µM) has a smaller molecular weight (217.28 Da), and it exhibited higher ACE inhibitory activity than KAF [19]. This may be attributed to the fact that Captopril, with its smaller molecular size, is more likely to enter the active pocket and inhibit ACE activity. Similarly, Captopril can enter the S1 active pockets according to the bind site Ala 354 according to the result of docking result (Table 2). Additionally, both KAF and Captopril can bind with ACE at several sites as follows: Arg522, Ala354, and Zn701 (Figure 4).

In order to validate the docking results, an MD simulation was conducted to analyze the dynamic behavior of complexes where atoms and molecules interact as a function of time [21]. Structural parameters including the root mean square deviation (RMSD), the root mean square fluctuation (RMSF), radius of gyration (Rg) and accessible surface area (SASA) were important factors when analyzing the flexibility and conformations of peptide–ACE complex in solution [22]. As shown in Figure 5a, the lower RMSD value of KAF-ACE exhibited a higher stability than Captopril-ACE before 20 ns. Similarly, the lower SAS value indicated the better stability of KAF-ACE before 10 ns. However, Captopril-ACE exhibited better stability according to RMSD, Rg and SAS value in plateau phase. RMSF calculates the fluctuations of each atom relative to its average position and is an indicator of the degree of freedom of atomic motion. In this study, there were five similar flexible areas (motion amplitude) (100–108; 150–161; 286–319; 430–445 ns and 496–520) in both KAF–ACE and Captopril–ACE complex (Figure 5c). Interestingly, these flexible regions were located far from the hydrogen bond sites between KAF and ACE (Asp415 and Ala354), which explains the affinity between the peptides and ACE.

### 2.5. Effect of Synthetic Peptides on NO Production, eNOS Activity in HUVECs

In vitro HUVEC incubation has been widely used to investigate the potential anti-hypertensive mechanism of drug-candidates [15]. In this study, KAF has no effect on HUVEC proliferation in the concentration of 0–100 µM (Figure 6a). NO, an endothelium-derived relaxing factor, is produced by eNOS in HUVECs. The phosphorylation of eNOS is known to enhance NO production, leading to vasodilation [23]. As shown in Figure 6b, KAF was found to enhance eNOS activity, resulting in increased NO production at a concentration of 100 µM. Previous studies have reported the involvement of several cytokines, such as Akt [24], AMP-activated protein kinase (AMPK) [25] and mitogen-activated protein kinase (P38, Jun) [26] in the phosphorylation of eNOS. In order to study explore the activation mechanism of peptides on eNOS. Some commercial protease inhibitors, such as Compound C (AMPK inhibitor), SB203580 (P38 inhibitor), LY294002 (Akt inhibitor), and SP600125 (Jun inhibitor), were used to interfere eNOS phosphorylation induced by peptide. Western blot shown that only LY294002 can prevent eNOS phosphorylation (*p* < 0.05) (Figure 7). This suggests that Akt may be involved in KAF-dependent eNOS phosphorylation. Furthermore, KAF treatment was found to significantly increase the intracellular Ca^2+^ content in HUVECs compared to the untreated peptide group (Figure 8). Ca^2+^ and calmodulin kinases were important in eNOS phosphorylation and NO production [27]. Previous studies have reported the existence of channels proteins and their function of increasing Ca^2+^ level in endothelial cells [28]. 

Additionally, the involvement of Ca^2+^/calmodulin kinases, such as CaMKII, in increasing intracellular Ca^2+^ levels and facilitating Ca^2+^/calmodulin-mediated eNOS activation has been reported [29]. In this study, calcium chelators (EGTA), CaMK-II inhibitor (KN93), LTCC blockade (Tetracaine) and RyR inhibition (Nifedipine) attenuated the KAF-induced eNOS phosphorylation, which indicated that Ca^2+^ and Ca^2+^ channels were also involved in KAF-induced eNOS phosphorylation (Figure 9). A similar situation has been reported in Betulinic acid, which activates eNOS phosphorylation and NO synthesis via the Ca^2+^/calmodulin-dependent protein kinase kinase/AMPK pathways [25]. Therefore, it can be concluded that KAF may activate LTCC and RyR, leading to an increase in intracellular Ca^2+^ levels. This rise in Ca^2+^ serves as a signaling molecule that promotes Akt-dependent eNOS phosphorylation and subsequent NO production. These results suggest that KAF has various beneficial effects on HUVECs.

## 3. Materials and Methods

### 3.1. Materials and Reagents

*U. prolifera* was obtained from the Institute of Oceanography, Chinese Academy of Sciences (Qingdao, China). Angiotensin I-converting enzyme (from rabbit lung), N-hippuril-L-histidy-L-leucine (HHL), Ang II, Neutral protease was purchased from Sangon Biotech (Shanghai, China). Sephadex G-100 were purchased from Auyoo Biotechnology Co. (Shanghai, China). Ultrafiltration membrane was purchased form Merck Millipore (Burlington, VT, USA). LY294002, SB203580, LY294002 SP600125, PD 98059, tetracaine, Compound C were purchased from Merck (Darmstadt, Germany). Nitric Oxide Assay Kit was purchased from Beyotime (Shanghai, China). MTT cell proliferation and cytotoxicity assay kits were purchased from Sigma-Aldrich (St. Louis, MO, USA). HUVEC cell line was purchased from Sangon Biotech (Shanghai, China). All other reagents were of analytical reagent grade.

### 3.2. Preparation of U. prolifera Protein Hydrolysate

The protein from *U. prolifera* was extracted, and the concentration of peptides was determined as described in our previous report [30]. The enzymatic hydrolysis condition accorded with the reported method with some modifications [11]. The protein was then dissolved in deionized water at a concentration of 10% (*w*/*v*). The pH and temperature of the solution were adjusted to the appropriate levels. Neutral proteases were added to the solution at a pH of 7.4, temperature of 47 °C, and an enzyme-to-substrate ratio of 3500 U/g (1% enzyme/substrate, *w*/*w* extracted protein powder). The hydrolysis reaction was carried out for 1 h and then terminated using a boiling water bath. The DH and ACE inhibitory activity of the *U. prolifera* protein hydrolysate was measured according to the previous report [11]. 

### 3.3. Determination of ACE Inhibitory Activity

The ACE inhibitory rate was explored according to a previous report [11]. Briefly, 80 μL of 5 mM HHL solution was mixed with 10 μL of peptide solution, followed by incubation for 5 min at 37 °C. Subsequently, 10 μL of ACE solution (0.1 U/mL) was added, followed by incubation at 37 °C for 60 min. The reaction was terminated by 200 μL HCl (1 M). In the blank group, peptide solution was replaced by 0.1 M sodium borate buffer. The reaction production was extracted with 1500 μL of ethyl acetate with slight oscillation for 1 min. Then, the mixture was centrifuged at 4000 rpm for 15 min, 1 mL of supernatant was transferred to another test tube, mixed with 1000 μL of acetic anhydride and 2000 μL of 0.5% (*v*/*v*) p-dimethyl amino benzaldehyde in pyridine, and then incubated at 40 °C for 30 min prior to spectrophotometric measurement at 459 nm. The degree of ACE inhibition (in percentage) was calculated according to Equation (1):The inhibition of ACE (%) = (Ab − Aa)/(Ab − Ac) × 100%(1)
where Aa represents the mixture of HHL, peptide and ACE; Ab represents the mixture of HHL and ACE; and Ac represents the mixture of HHL and inactive ACE. The IC50 value was defined as the inhibitor concentration inhibiting 50% activity of ACE.

### 3.4. Purification and Identification of ACE-Inhibitory Peptides from Hydrolysates

The hydrolysates were first purified by Sephadex-G100 column (2.5 cm × 70 cm) with deionized water as the elution at a flow rate of 1.0 mL/min. Fractions were collected at 2 min intervals, and each fraction was tested for ACE inhibitory activity. The fraction with the highest ACE inhibitory activity was further fractionated using an ultrafiltration membrane, resulting in three fractions: <3 kDa, 3–10 kDa, and >10 kDa. The purified fractions were then analyzed using online nano-flow liquid chromatography tandem mass spectrometry, following the methods described in our previous report [30].

### 3.5. Screening and Synthesis of the Potential ACE Inhibitory Peptides

The identified peptides were determined by BIOPEP (https://biochemia.uwm.edu.pl/biopep-uwm/, accessed on 15 July 2022) to remove the reported peptides [4]. The peptides were then evaluated for various biological parameters such as biological activity potential, solubility, toxicity, and human intestinal absorption (HIA) using Peptide Ranker, Innovagen, and admetSAR (ecust.edu.cn; accessed on 15 July 2022) tools, respectively. Additionally, the affinities between the selected peptides and ACE were assessed through molecular docking using Discovery Studio 2022 software with the ACE crystal structure (ID: 1O8A) chosen as the receptor. Finally, the selected peptide was synthetized by China peptides (Shanghai, China) (peptide purity 98%).

### 3.6. Inhibition Kinetics

The kinetics of the peptide on ACE was explored using the Lineweaver−Burke plot. The protocol was the same as in the ACE inhibition assay. The experimental procedure was similar to the ACE inhibition assay. ACE was preincubated with peptides of different concentrations ranging from 0 to 100 µM, and then reacted with hippuryl-L-histidyl-L-leucine (HHL) at various concentrations ranging from 1 to 5 mM. The Michaelis–Menten constant (Km) and maximum initial velocity (Vmax) were calculated based on the y- and x-axis intercepts of the primary plot, respectively.

### 3.7. Molecular Dynamics (MD) Simulation

To validate the docking results, a 50 ns MD simulation was conducted for ACE in complex with selected peptide and the control group (Captopril). The MD simulation was performed using the GROMACS program (version 2020) and the Amber ff12SB force field, following a previously reported protocol [31].

### 3.8. Cell Culture of HUVEC

The cytotoxicity assay of HUVECs was performed using the MTT Cell Proliferation and Cytotoxicity Assay Kit [32]. NO production and eNOS activity in HUVECs treated with peptides were measured by Total Nitric Oxide Assay Kit and Nitric Oxide Synthase Assay Kit. HUVECs with a content of 1 × 10^6^ cells /bottle were seeded in T25 cell cultured bottle for 24 h. Subsequently, the cells were treated with peptides at 100 µM for 24 h. After that, the cell culture medium containing peptides and specific protease inhibitors at appropriate final concentrations (EGTA, 500 nM; KN93, 10 µM; Nifedipine, 60 µM; Tetracaine, 60 µM; Compound C, 10 µM; SB203580, 10 µM; LY294002, 10 µM; SP600125, 10 µM) were used to culture HUVECs for additional 6 h [25].

### 3.9. Statistical Analysis

All experiments were performed in triplicate and the results reported as the mean ± standard deviation (SD). Data were analyzed by one-way analysis of variance (ANOVA), and then Dunnett multiple tests were performed using GraphPad Prism Version 9 (San Diego, CA, USA). Significance level was set at *p* less than 0.05.

## 4. Conclusions

In the above-presented work, a novel anti-ACE peptide KAF (0.63 ± 0.26 µM) was separated and identified from *U. prolifera* protein. This peptide effectively inhibits ACE activity through competitive bind, primarily due to the formation of two conventional hydrogen bonds. In HUVECs, KAF can activate the eNOS activity in the generation of NO. KAF can increased the intracellular Ca^2+^ level through the LTCC and the RyR in endoplasmic reticulum, and completed the activation of eNOS under the mediation of Akt signaling pathway. These findings supported that KAF has the potential to be a safe candidate against ACE and relax blood vessel. Subsequently, the antihypertension activity of KAF on hypertension rat will be performed.

## Figures and Tables

**Figure 1 marinedrugs-22-00398-f001:**
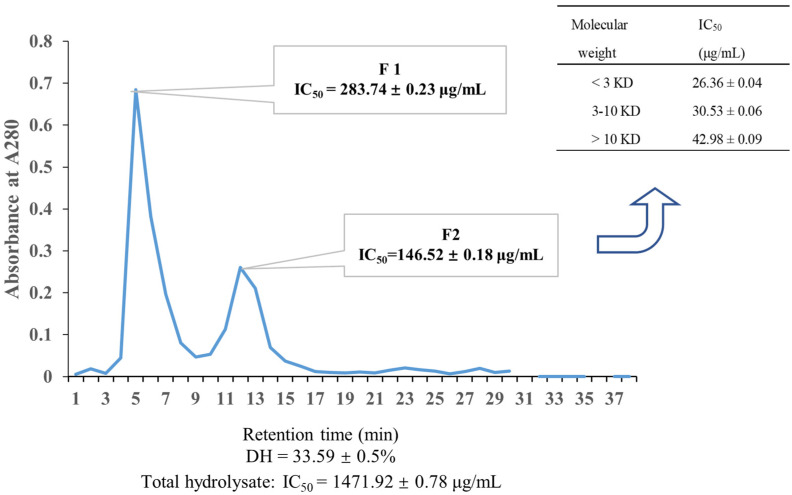
Purification of ACE inhibitory peptides. Sephadex G-100 gel chromatography of fractions from hydrolysates (neutral protease); the IC_50_ value of different fractions from the ultrafiltration of Fraction 2 (F2) are exhibited in the inserted table. Data are presented as mean ± SD.

**Figure 2 marinedrugs-22-00398-f002:**
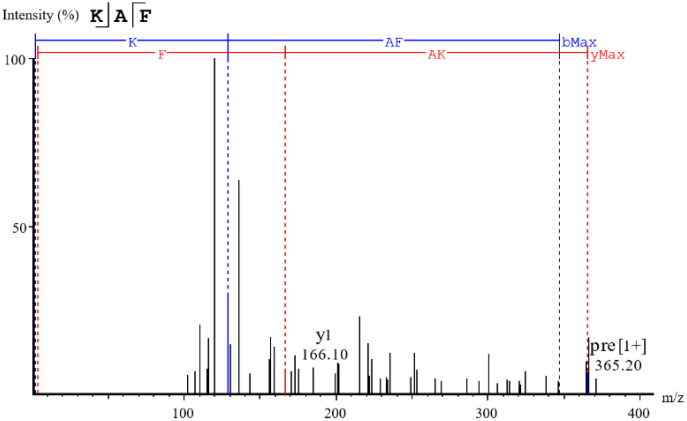
MS/MS spectra of the selected peptide.

**Figure 3 marinedrugs-22-00398-f003:**
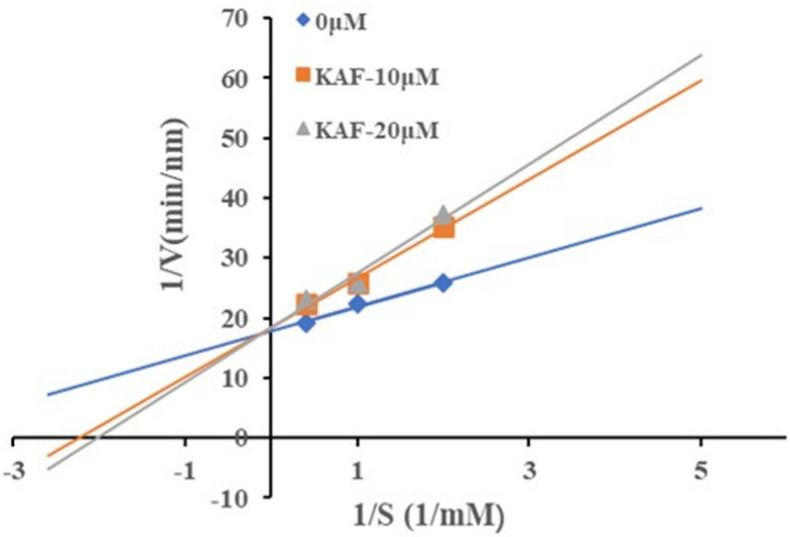
Lineweaver–Burk plots of ACE inhibition by KAF.

**Figure 4 marinedrugs-22-00398-f004:**
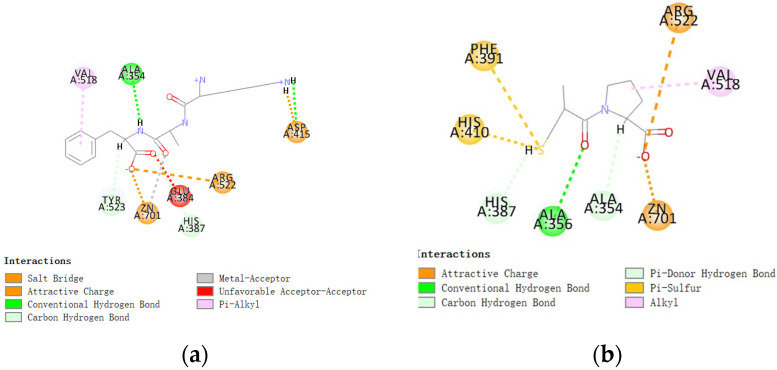
Docking simulation of candidates with ACE. Two-dimensional interaction of KAF–ACE (**a**) and Captopril–ACE (**b**).

**Figure 5 marinedrugs-22-00398-f005:**
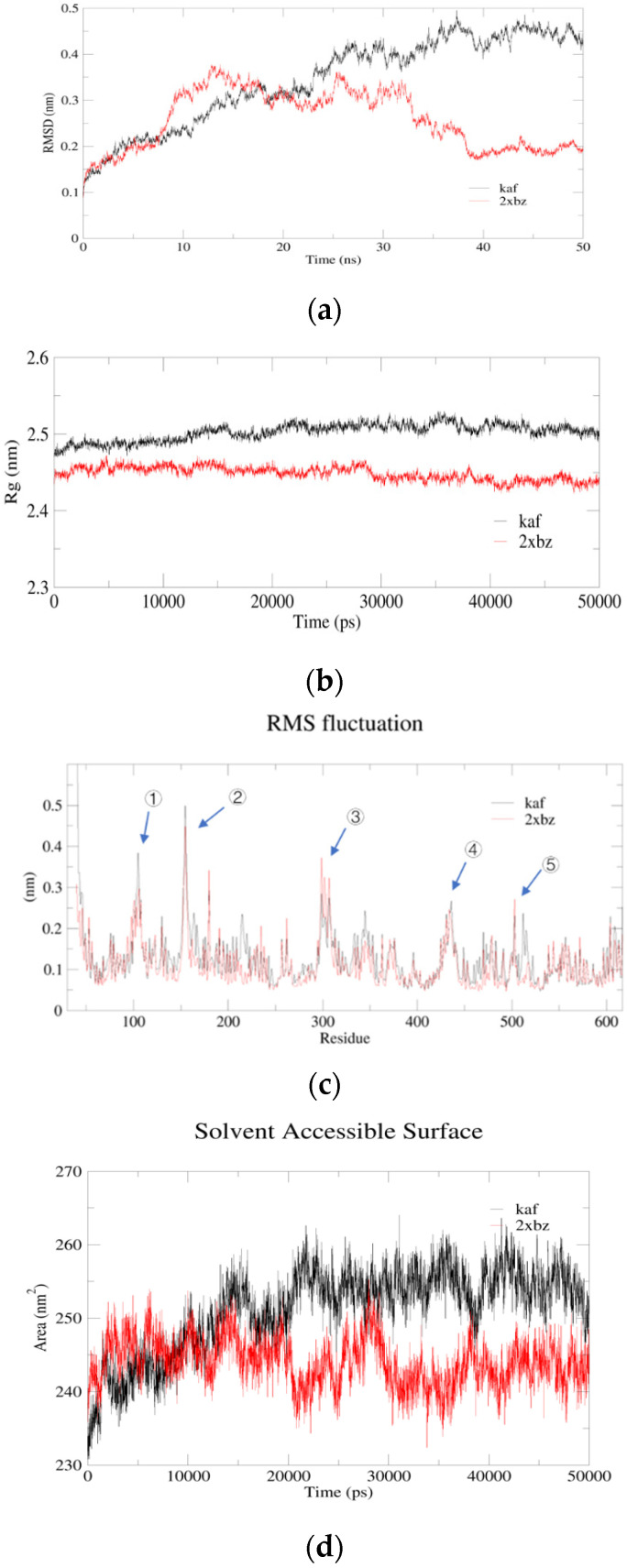
The result of molecular dynamics simulation (kaf: KAF; 2xbz: Captopril; (**a**) RMSD; (**b**) Rg; (**c**) RMSF (the number represents of flexible areas); (**d**) SASA).

**Figure 6 marinedrugs-22-00398-f006:**
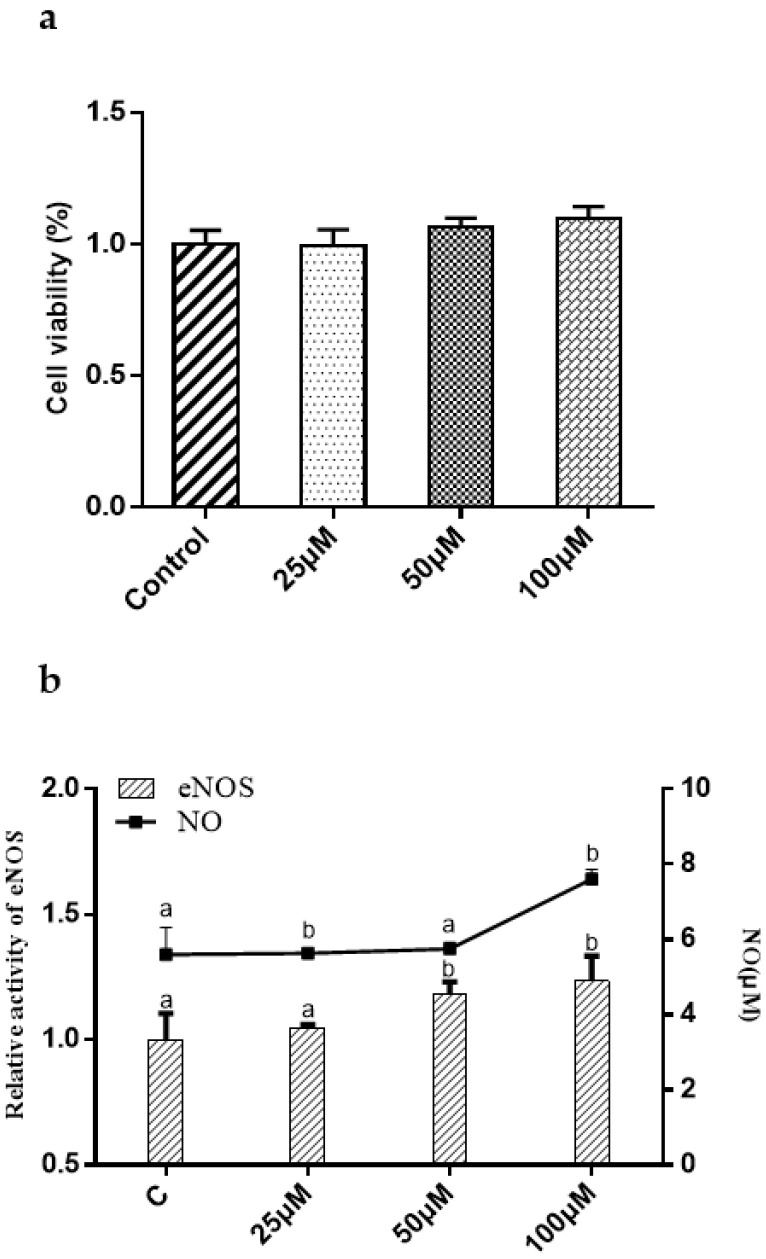
Effect of this *Ulva prolifera* peptide (KAF) on HUVECs. (**a**) Effect of peptide on HUVECs viability; (**b**) effect of peptide on eNOS activity and NO production in HUVECs. The cells were treated with peptides at 0, 25, 50, and 100 µM for 24 h; values that do not share a common superscript lowercase letter within a column differ significantly, *p* < 0.05.

**Figure 7 marinedrugs-22-00398-f007:**
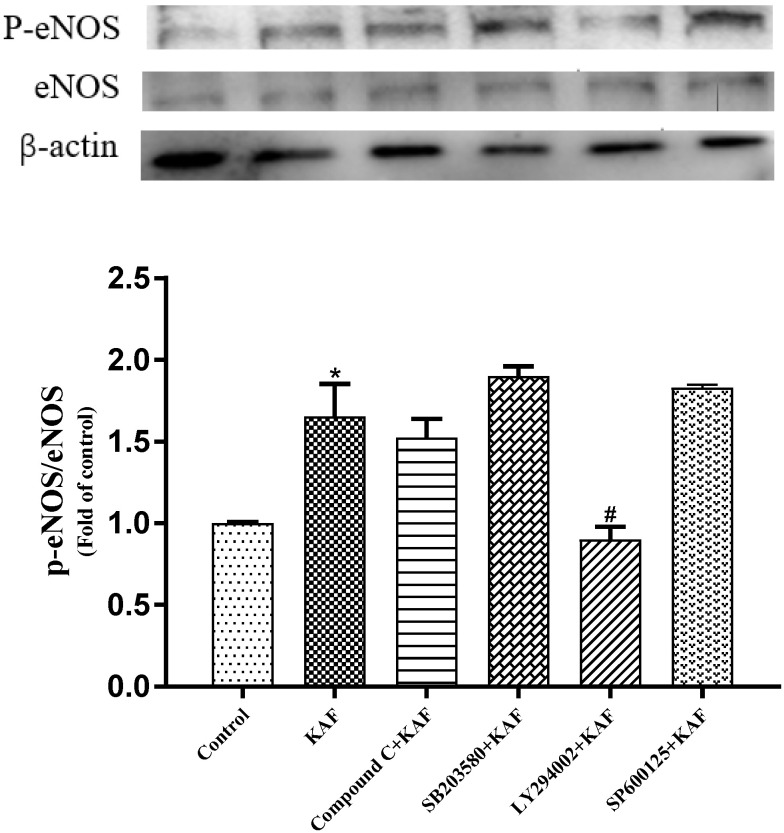
The effect of *U. prolifera* peptide (KAF) *on* eNOS phosphorylation. Control, Cells were cultured by medium for 24 h; KAF, cells were cultured by 100 µM KAF for 18 h; Compound C (AMPK inhibitor) + KAF, cells were cultured by 100 µM KAF for 12 h and then, adding 10 µM Compound C for another 6 h; SB203580 (P38 inhibitor) + KAF, cells were cultured by 100 µM KAF for 12 h and then, adding 10 µM SB203580 for another 6 h; LY294002 (Akt inhibitor) + KAF, cells were cultured by 100 µM KAF for 12 h and then, adding 10 µM LY294002 for another 6 h; SP600125 (Jun inhibitor) + KAF, cells were cultured by 100 µM KAF for 12 h and then, adding 10 µM SP600125 for another 6 h; * *p* < 0.05, compared with control; # *p* < 0.05, compared with KAF treatment.

**Figure 8 marinedrugs-22-00398-f008:**
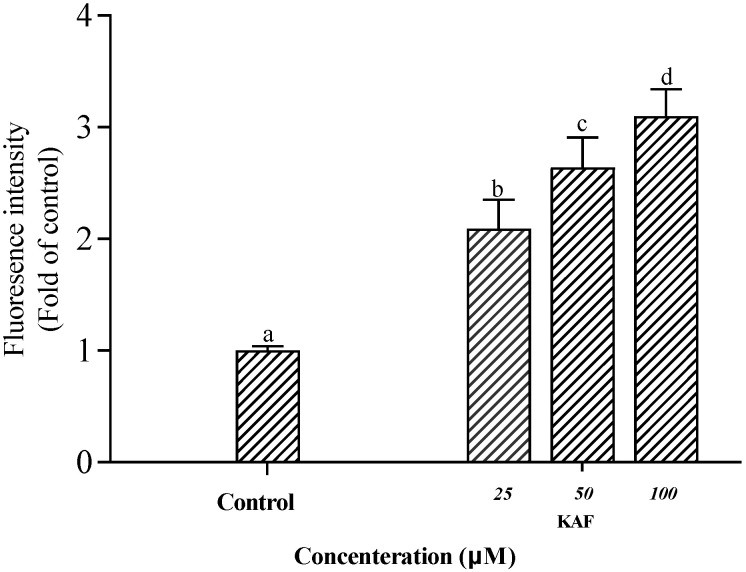
Effect of *U. prolifera* peptide (KAF) on Ca^2+^ content in HUVECs (values that do not share a common superscript lowercase letter within a column differ significantly, *p* < 0.05).

**Figure 9 marinedrugs-22-00398-f009:**
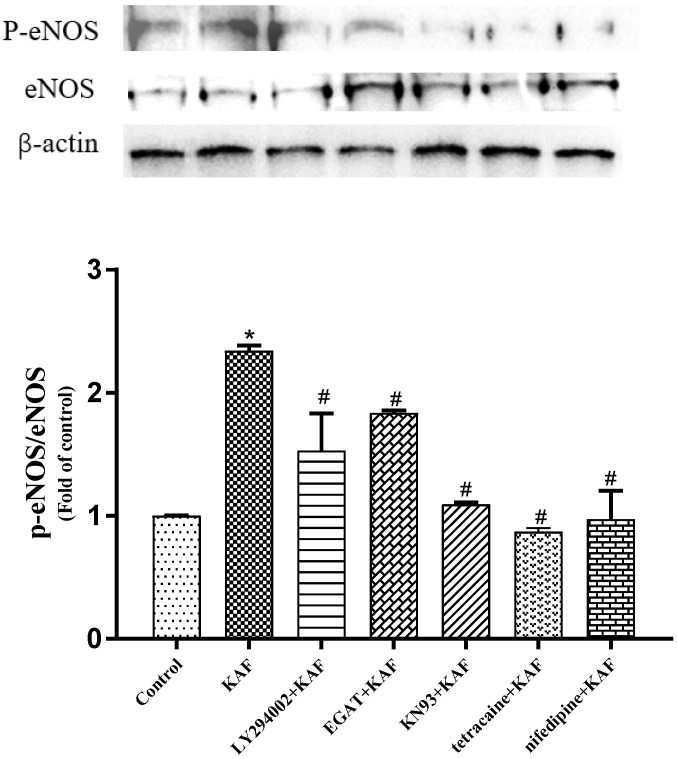
Role of Ca^2+^ channels on *Ulva prolifera* peptide (KAF)-induced eNOS phosphorylation. Control, Cells were cultured by medium for 24 h; KAF, cells were cultured by 100 µM KAF for 18 h; LY294002 (Akt inhibitor) + KAF, cells were cultured by 100 µM KAF for 12 h and then, adding 10 µM LY294002 for another 6 h; EGTA (calcium chelators) + KAF, cells were cultured by 100 µM KAF for 12 h and then, adding 500 nM EGTA for another 6 h; KN93 (CaMK-II inhibitor) + KAF, cells were cultured by 100 µM KAF for 12 h and then, adding 10 µM KN93 for another 6 h; Tetracaine (LTCC blockade) + KAF, cells were cultured by 100 µM KAF for 12 h and then, adding 60 µM Tetracaine for another 6 h; Nifedipine (RyR inhibition) + KAF, cells were cultured by 100 µM KAF for 12 h and then, adding 60 µM Nifedipine for another 6 h; ***** *p* < 0.05, compared with control; # *p* < 0.05, compared with KAF treatment.

**Table 1 marinedrugs-22-00398-t001:** Potential ACE inhibitory peptides form protease hydrolysis.

Peptide	Peptide Ranker	WS	Toxin	HIA	BBB	-C Dock Energy	IC_50_ (μM)
DRW	0.88931	GOOD	NO	0.7047	−0.8101	fail	
GMR	0.884249	GOOD	NO	0.6327	−0.6948	fail	
MGR	0.860347	GOOD	NO	0.6327	−0.6948	fail	
RYFR	0.846704	GOOD	NO	0.4488	0.8791	fail	
KWY	0.794787	GOOD	NO	0.9434	−0.8790	fail	
RWK	0.774831	GOOD	NO	0.8444	−0.8040	fail	
LGSFR	0.737369	GOOD	NO	0.8291	−0.7994	fail	
EGRW	0.734044	GOOD	NO	0.8291	−0.7994	fail	
WRAA	0.702731	GOOD	NO	0.9124	−0.8063	fail	
KAF	0.682106	GOOD	NO	0.7042	−0.8181	88.967	0.63 ± 0.26
DFT	0.600067	GOOD	NO	0.4747	−0.8839	fail	
ERFY	0.464208	GOOD	NO	0.6018	−0.8872	fail	
PAMK	0.419096	GOOD	NO	0.774	−0.7314	fail	
Captopril						46.94	0.017 [19]

Note: Peptide ranker, >0.5 represents of high biological activity; WS, water solubility; Toxin, water-solubility HIA+, high human intestinal absorptivity (>30%); BBB+, high blood–brain barrier permeability (>0.1); -CE score, the score of -C Docker energy (kcal/moL). Fail represents of docking failure between candidates and ACE. Data are presented as mean ± SD.

**Table 2 marinedrugs-22-00398-t002:** Interaction information.

Candidates	Bond Position	Distance(Å)	Type	Number
KAF	A: ASP415:OD2-KAF:H22 A: ARG522:NH1-KAF: O44 A: ZN701: ZN-KAF: O44	2.16146 4.49144 2.24177	Attractive Charge	3
	A: ASP415:OD1-KAF: H21 A: ALA354:O-KAF: H36	2.26404 2.3907	Conventional Hydrogen Bond	2
	A: HIS387:HD2-KAF: O44 A: TYR423: OH-KAF: H38	3.01083 2.3496	Carbon Hydrogen Bond	2
	A: VAL418-KAF	4.90944	Pi-Alkyl	1
	A: ZN701: ZN-KAF: O34	3.0144	Metal-Acceptor	1
Captopril (Cap)	A: ARG522:NH1-Cap: O11 A: ZN701: ZN-Cap: O11	4.49391 2.27274	Attractive Charge	2
	A: ALA356:HN-Cap: O1	2.89368	Conventional Hydrogen Bond	1
	A: ALA354:O-Cap: H22	2.91634	Carbon Hydrogen Bond	1
	A: HIS387-Cap:H18	2.76629	Pi-Donor Hydrogen Bond; Pi-Sulfur	1
	A: VAL518-Cap	4.64702	Alkyl	1
	A: PHE391-Cap: S4 A: HIS410-Cap: S4	4.77799 4.40347	Pi-Sulfur	2

## Data Availability

The data presented in this study are available on request from the corresponding author.

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
