# Peer review of "Preparation and Vasodilation Mechanism of Angiotensin-I-Converting Enzyme Inhibitory Peptide from Ulva prolifera Protein"

_marinedrugs, 2024, doi:10.3390/md22090398_

Round 1

Reviewer 1 Report

Comments and Suggestions for Authors

I did not found any serious mistakes,   but for future work

I would recommend to test domain specificity of KAF peptide.

You used HHL which cleaved in the somatic (rabbit lung) ACE

mainly by C domain active center  (Wei, 1991).

The solutions are

1) to use testicular ACE as a representative of C domain ACE (Franke, 

2) to use ileal ACE (Deddish. 1994) as a representative of N domain 

or alternatively to use both substrate- HHL (which was cleaved mainly by C domain active center) and ZPHL, which cleaves by both domain active center, (Danilov, 2008)

Author Response

Comments and Suggestions for Authors

I did not found any serious mistakes, but for future work

I would recommend to test domain specificity of KAF peptide.

You used HHL which cleaved in the somatic (rabbit lung) ACE

mainly by C domain active center (Wei, 1991).

The solutions are

1) to use testicular ACE as a representative of C domain ACE (Franke, 

2) to use ileal ACE (Deddish. 1994) as a representative of N domain 

or alternatively to use both substrate- HHL (which was cleaved mainly by C domain active center) and ZPHL, which cleaves by both domain active center, (Danilov, 2008)

Response: Many ACE-inhibitory peptides were screened by the reaction of HHL and ACE (rabbit lung), such as

ITAPHW (https://doi.org/10.1016/j.foodchem.2023.137921),

 VGLFPSRSF (https://doi.org/10.1016/j.foodchem.2023.137074)

and SAGGYIW(https://doi.org/10.1016/j.jff.2019.103751).

We agree with this comment. Thank you for your reminding and we will improve the reaction according to your suggestion in the further study. 

Reviewer 2 Report

Comments and Suggestions for Authors

Abstract

What does ADMET meaning?

Introduction

Please add action mechanisms of ACE inhibitor

Could you explain the cellular mechanism by which the peptides regulate the pressure?

Line 32. After the quotation [3] check whether it is “;” or “.”

Line 39. Delete “with economic benefits”.

Results and discusión

In general, I think that a more in-depth discussion of the results that have been obtained in this part is necessary.

Lines 71- 75. Please check the redaction and improve it.

Line 75 Please add the optimal conditions of U. prolifera protein hydrolisis and also the reference

Line 77-78 Please explain why do you use sephadex-G100.

Line 111 What does HHL means?.

Line 83. Mention that the smaller peptides have greater activity. But don't explain why.

Mention that captopril is a small molecule but do not include the bibliographic citation.

Line 112. The KAF peptide is a competitive competitor. In relation to antihypertensive activity, explain whether this type of inhibition is beneficial or not.

Line 118. Molecular docking typically involves normalization of the protein of interest. This means that hydrogens are added, water is removed, and if the protein has ligands, these are also removed. The PDBQT file of the protein is checked and the minerals present (if the protein has them) are removed to make the docking as "natural" as possible. Check the docking and improve the quality of the images (it is very hard to see).

Line 139. Similarly, if the docking is modified, the molecular dynamics will be modified. Please improve this part.

Author Response

Comments and Suggestions for Authors

we are grateful for the insightful comments on our paper. The revised text has been marked in blue.     

Abstract

What does ADMET meaning?

Response:  ADMET consists of Absorption, Distribution, Metabolism and Excretion and Toxicity. ADMET is used to evaluate the drug properties of small molecules and it has been used to screen and evaluate the ACE inhibitory peptides (https://doi.org/10.1016/j.foodchem.2021.131070).

Introduction

Please add action mechanisms of ACE inhibitor

Response: Action mechanisms of ACE inhibitor has been added at Line 36.

Could you explain the cellular mechanism by which the peptides regulate the pressure?

Response: Peptides can inhibit the activity of ACE to prevent the production of Ang II (Vasoconstrictor) and activate eNOS activity to produce NO (Vasodilator).

Line 32. After the quotation [3] check whether it is “;” or “.”

Response: “;” is more suitable for expressing the coherence of the preceding and following sentences according to the suggestion of the professional company. Line34, 95.

Line 39. Delete “with economic benefits”.

Response: “with economic benefits” has been deleted according to your suggestion. Line 42

Results and discusión

In general, I think that a more in-depth discussion of the results that have been obtained in this part is necessary.

Lines 71- 75. Please check the redaction and improve it.

Response: Thank you for your reminding. Citations have been corrected to apply to the magazine. Line 71-77

Line 75 Please add the optimal conditions of U. prolifera protein hydrolisis and also the reference

Response: the optimal conditions and reference has been added and marked in blue.

“Neutral proteases were added to the solution at a pH of 7.4, temperature of 47°C, and an enzyme-to-substrate ratio of 3500 U/g (1% enzyme/substrate, w/w extracted protein powder)” Line 250-252

Line 77-78 Please explain why do you use sephadex-G100.

Response: Sephadex has been widely used to separate peptides from enzyme hydrolysis (http://dx.doi.org/10.1016/j.foodchem.2016.05.087). The separation range of Sephadex-G100 is 4000-15000 Da and it can exclude undecomposed proteins. Subsequently, ultrafiltration is used to further separate smaller components in this study.

The description of sephadex-G100 has been added at Line 84.

Line 111 What does HHL means?.

Response: Hippuryl-His-Leu-OH (HHL) is the substrate of ACE. The His Leu released by HHL can react with ortho benzaldehyde or fluorescein amine (HY-D0715) and this reaction is used for ACE activity detection.

Line 83. Mention that the smaller peptides have greater activity. But don't explain why.

Mention that captopril is a small molecule but do not include the bibliographic citation.

Response: The reason for smaller peptides has greater activity has been added.

“These findings support the notion that smaller peptides are more likely to enter the active center and possess stronger ACE inhibitory activity” Line 89

The citation of captopril has been added (https://doi.org/10.1021/acs.jafc.5b05869) Line 135

Line 112. The KAF peptide is a competitive competitor. In relation to antihypertensive activity, explain whether this type of inhibition is beneficial or not.

Response: Competitive Inhibitors and enzyme substrates have similar structures and binding sites, and they compete with each other for enzyme binding. It is one of reversible inhibitors molecules do not undergo chemical reactions with amino acid residues at the enzyme active site. With the removal of inhibitory effects, the function of enzymes can be restored. Hence, KAF will not disrupt the structure of the ACE and has lower side effects.

Line 118. Molecular docking typically involves normalization of the protein of interest. This means that hydrogens are added, water is removed, and if the protein has ligands, these are also removed. The PDBQT file of the protein is checked and the minerals present (if the protein has them) are removed to make the docking as "natural" as possible. Check the docking and improve the quality of the images (it is very hard to see).

Response: Thank you for you reminding. There revised model (ACE) has no water or minerals and the docking conditions are according to the pervious report (http://dx.doi.org/10.1016/j.foodchem.2016.05.087). The images have been changed to vector diagram (.png style). Line 141

Line 139. Similarly, if the docking is modified, the molecular dynamics will be modified. Please improve this part.

Response: We conducted multiple molecular docking and the results were consistent.

Reviewer 3 Report

Comments and Suggestions for Authors

In this manuscript, the authors reported the discovery of an anti-ACE peptide from Ulva prolifera. The authors adopted an in vitro-in silico integrated approach, using techniques including protein expression analysis, cellular assays, molecular docking and dynamics simulation, and enzyme kinetic analysis. I have some feedback as listed below.

1.     Please recheck for typo and grammatical errors in the whole text. For example, in line 32, the word after the semi-colon mark begins with an uppercase letter.

2.     Title can be made more concise/accurate by revising to “Preparation and Vasodilation Mechanism of ACE Inhibitory Peptide from Ulva prolifera Protein

3.     ABSTRACT – lacking a concluding statement. Furthermore, abbreviations are used only once in most cases, thus their introduction in the ABSTRACT seems unnecessary.

4.     INTRO:

           (i).         Please recheck the whole text to ensure that species names are in italics. For example, see line 36: “Chlorella vulgaris”.

          (ii).         Lines 48-50: Wrong reference. Ref #10 is not about U. prolifera as claimed by the authors. Please recheck and rectify.

        (iii).         Lines 53-60: This whole block of text should be supported by at least a relevant reference.

        (iv).         After reading the INTRO, it is unclear what the novelty is for this manuscript. Discovery of anti-ACE peptides from U. prolifera proteins hydrolyzed by neutral protease has been previously reported (Li et al. (2023) Purification identification and function analysis of ACE inhibitory peptide from Ulva prolifera protein. Food Chemistry, 401.) So, the authors should make it clear in INTRO how this current work is novel enough to deserve publication.

5.     RESULTS & DISCUSSION:

           (i).         Overall, the section lacks in-depth/insightful interpretation and discussion of the data presented in relation to what have been published in the literature. This compromises the significance of this current study.

          (ii).         Lines 73-74: Wrong reference. Ref #10 is not about U. prolifera as claimed by the authors. Please recheck and rectify.

        (iii).         Line 76: The word “fortunately” seems inappropriate to use here. Please revise.

        (iv).         Line 81: The statement that the < 3 kDa fraction showed the highest activity seems to be subjective. The IC50 data, as shown in the table in Fig 1, should be substantiated by statistical analysis.

          (v).         Fig 1: Unclear why the fractions were monitored at 280 nm but not other wavelengths also commonly used e.g., 220 nm.

        (vi).         The reason for analyzing DH is unclear. There is no discussion on it at all.

       (vii).         Table 1: For captopril, is there a typo error or the authors only performed one replicate of measurement for its activity? If so, this should be repeated two more times, and a standard deviation value should be calculated and shown.

     (viii).         Fig 3: Lineweaver-Burk plots - how many reps were done? At least three reps are needed.

        (ix).         Fig 5: MD results - how many reps were done? At least three reps are needed. Also, please improve the caption as it is not informative in its current form. For example, after reading the caption, it is unclear what the numbers 1-5 in chart C refer to. Also, the charts could have been enlarged to make it more legible considering there is so much unused space around the figure. The font size is too small in the charts. This issue applies to other figures too.

          (x).         Figs 7 & 9: The caption says “*P<0.5, compared with control; #p<0.5, compared with KAF treatment.” – There are two problems here. The authors used both uppercase (P) and lowercase (p) when indicating probability. Please standardize it. But more importantly, shouldn’t the probability levels be set to 0.05 instead of 0.5? Please take note that if it is 0.5, it is unacceptable, and the analysis should be repeated.

        (xi).         Fig 8: Please correct the typo in the y-axis title: “Flod”. More importantly, it is more appropriate to use multiple comparison statistical test here to analyze the statistical significance between the fluorescence intensity data for 25, 50, and 100 uM KAF. That way, the dose-dependent trend can be confirmed.

6.     M&M

           (i).         Line 238: It is unclear why 10% protein was deemed appropriate. Did the authors have any data to show that this is appropriate?

          (ii).         Line 240: “1% enzyme/substrate, w/w protein” – It is unclear as well how 1% was determined to be appropriate. Also, was the protein mass here referring to the mass of protein extract/isolate? Or based on protein content as determined by a biochemical assay?

        (iii).         Line 246: Sephadex-G100 seems not a suitable resin to use when fractionating peptides, but might be more useful for separating proteins. Why was this resin used?

        (iv).         Lines 254-255: “The identified peptides were determined by BIOPEP (https://bio- chemia.uwm.edu.pl/biopep-uwm/) to remove the reported peptides.” – This statement is unclear/confusing. Importantly, when using any online databases, the access date should be indicated. Also, please cite the paper for BIOPEP as per requested by the authors.

          (v).         For docking, the crystal used by the authors was 1o8A. This is a crystal for the human testicular ACE. Why didn’t the authors use the crystal for human somatic ACE, e.g., 4APJ? The somatic ACE is much more prevalent than the testicular ACE after all?

        (vi).         Line 277: Please check whether it should be “NO” or “No”.

       (vii).         3.8 Statistical Analysis: The number of replicates is not clearly indicated. The statement “Significant level was set at 0.05” is unclear too.

Author Response

Review 3

Comments and Suggestions for Authors

In this manuscript, the authors reported the discovery of an anti-ACE peptide from Ulva prolifera. The authors adopted an in vitro-in silico integrated approach, using techniques including protein expression analysis, cellular assays, molecular docking and dynamics simulation, and enzyme kinetic analysis. I have some feedback as listed below.

Response: We appreciate the time and effort that you dedicated to providing feedback on our manuscript. The revised text has been marked in red.

Question 1. Please recheck for typo and grammatical errors in the whole text. For example, in line 32, the word after the semi-colon mark begins with an uppercase letter.

Response: The word “This” has been corrected to “this” according to your suggestion. (Line 34, 95).

Question 2. Title can be made more concise/accurate by revising to “Preparation and Vasodilation Mechanism of ACE Inhibitory Peptide from Ulva prolifera Protein

Response: We agree with this comment. The title of this manuscript has been changed according to your suggestion. (Line 2)

Question 3.     ABSTRACT – lacking a concluding statement. Furthermore, abbreviations are used only once in most cases, thus their introduction in the ABSTRACT seems unnecessary.

Response: Thank you for your suggestion. A concluding statement has been supplied in ABSTRACT (Line 23-24). Besides, abbreviations “PI3K/Akt” in ABSTRACT has been deleted (Line 22, 25, 297) and other abbreviations (ACE, eNOS, NO, Ca2+, LTCC, RyR) are used more than three times in the text.

Question 4.     INTRO:

           (i).         Please recheck the whole text to ensure that species names are in italics. For example, see line 36: “Chlorella vulgaris”.

Response: The species names of “Chlorella vulgaris” has been corrented. (Line 39). Other species names have been checked.

          (ii).         Lines 48-50: Wrong reference. Ref #10 is not about U. prolifera as claimed by the authors. Please recheck and rectify.

Response: Enteromorpha clathrate (marked in Ref #10) is the original name of U. proliferate.

        (iii).         Lines 53-60: This whole block of text should be supported by at least a relevant reference.

Response: Some references have been added (Line 60, 63).

        (iv).         After reading the INTRO, it is unclear what the novelty is for this manuscript. Discovery of anti-ACE peptides from U. prolifera proteins hydrolyzed by neutral protease has been previously reported (Li et al. (2023) Purification identification and function analysis of ACE inhibitory peptide from Ulva prolifera protein. Food Chemistry, 401.) So, the authors should make it clear in INTRO how this current work is novel enough to deserve publication.

 Response: Thank you for your suggestion. The ACE-inhibitory peptide KAF from Ulva prolifera is a newly discovered. Besides, its vasodilation mechanism has not been reported before and the work is different with the reported (Li et al. (2023). We have revised the description to highlight the novelty of the existing work according to your suggestion. (Line 52-53)

  1. RESULTS & DISCUSSION:

           (i).         Overall, the section lacks in-depth/insightful interpretation and discussion of the data presented in relation to what have been published in the literature. This compromises the significance of this current study.

Response: Thank you for your comments. We will further improve the results and discussion section based on your suggestions

   (ii).         Lines 73-74: Wrong reference. Ref #10 is not about U. prolifera as claimed by the authors. Please recheck and rectify.

Response: Enteromorpha is now considered to be a synonym of Ulva (https://doi.org/10.1080/1364253031000136321). Enteromorpha spp is one kind of U. proliferate. We have revised the description of this type of algae Enteromorpha spp. (one of U. prolifera). Line 48

        (iii).         Line 76: The word “fortunately” seems inappropriate to use here. Please revise.

Response: The word has been corrected. “Then, a hydrolysate with ACE inhibitory activity was obtained” Line 78.

        (iv).         Line 81: The statement that the < 3 kDa fraction showed the highest activity seems to be subjective. The IC50 data, as shown in the table in Fig 1, should be substantiated by statistical analysis.

Response: The IC50 value of different fractions from the ultrafiltration of Fraction 2 (F2) were exhibited in the inserted table (Fig1). Among of the three fractions, the fraction (< 3 kDa) showed the lowest IC50 value.   

  (v).         Fig 1: Unclear why the fractions were monitored at 280 nm but not other wavelengths also commonly used e.g., 220 nm.

Response: The UV absorption wavelength of the polypeptide is in the range of 200-300 nm. Meanwhile, the wavelength of 280 nm was selected based on the reported literature (http://dx.doi.org/10.1016/j.foodchem.2016.05.087).

        (vi).         The reason for analyzing DH is unclear. There is no discussion on it at all.

Response: DH is a parameter that describes the degree of enzymatic hydrolysis. The analyzing of DH has been added.

 “DH is a parameter that describes the degree of U. prolifera protein hydrolysis, which is higher than the previous report[10]. The result indicates that a large amounts of peptides and amino acids were released from U. prolifera protein.” Line 79-81

       (vii).         Table 1: For captopril, is there a typo error or the authors only performed one replicate of measurement for its activity? If so, this should be repeated two more times, and a standard deviation value should be calculated and shown.

Response: Captopril is a marketed ACE inhibitor drug (IC50, 0.025 μM). We have inserted the correct references in Table 1. Line 97, 111

     (viii).         Fig 3: Lineweaver-Burk plots - how many reps were done? At least three reps are needed.

Response: The inhibition experiments of peptides were conducted three times under different concentration conditions.

        (ix).         Fig 5: MD results - how many reps were done? At least three reps are needed. Also, please improve the caption as it is not informative in its current form. For example, after reading the caption, it is unclear what the numbers 1-5 in chart C refer to. Also, the charts could have been enlarged to make it more legible considering there is so much unused space around the figure. The font size is too small in the charts. This issue applies to other figures too.

Response: Thank you for your suggestion. We conducted multiple molecular docking and the results were the same. The caption has been revised (Line 171) and a detailed description has been added in the text (Line 155). Clear images will be submitted in the later stage.

          (x).         Figs 7 & 9: The caption says “*P<0.5, compared with control; #p<0.5, compared with KAF treatment.” – There are two problems here. The authors used both uppercase (P) and lowercase (p) when indicating probability. Please standardize it. But more importantly, shouldn’t the probability levels be set to 0.05 instead of 0.5? Please take note that if it is 0.5, it is unacceptable, and the analysis should be repeated.

Response: Thank you for your attention. The error description of P value has been corrected. Line 205, 230

        (xi).         Fig 8: Please correct the typo in the y-axis title: “Flod”. More importantly, it is more appropriate to use multiple comparison statistical test here to analyze the statistical significance between the fluorescence intensity data for 25, 50, and 100 uM KAF. That way, the dose-dependent trend can be confirmed.

 Response: We agree with your point. The “Flod” in Fig 8 has been corrected. The statistical test of Fig 8 has been revised according to your suggestion. Line 207

  1. M&M

           (i).         Line 238: It is unclear why 10% protein was deemed appropriate. Did the authors have any data to show that this is appropriate?

 Response: 10% protein comes from the results of preliminary experiments. There may be more suitable conditions to obtain more peptides. The protein percent for enzyme hydrolysis will be explored in the further study.

          (ii).         Line 240: “1% enzyme/substrate, w/w protein” – It is unclear as well how 1% was determined to be appropriate. Also, was the protein mass here referring to the mass of protein extract/isolate? Or based on protein content as determined by a biochemical assay?

Response: “1% enzyme/substrate, w/w protein” is the result of our preliminary experiment.

Extracted protein powder was hydrolyzed to produce peptide. The protein name was revised (Line 252)

        (iii).         Line 246: Sephadex-G100 seems not a suitable resin to use when fractionating peptides, but might be more useful for separating proteins. Why was this resin used?

Response: Sephadex has been widely used to separate peptides from enzyme hydrolysis. (http://dx.doi.org/10.1016/j.foodchem.2016.05.087). The separation range of Sephadex-G100 is 4000-15000 and it can exclude undecomposed proteins. The discussion for Sephadex-G100 has been added.  Line 84

        (iv).         Lines 254-255: “The identified peptides were determined by BIOPEP (https://bio- chemia.uwm.edu.pl/biopep-uwm/) to remove the reported peptides.” – This statement is unclear/confusing. Importantly, when using any online databases, the access date should be indicated. Also, please cite the paper for BIOPEP as per requested by the authors.

Response: We have added the Access time: 2022.07 and citation at the end of BIOPEP address according to your suggestion. Line 266.

          (v).         For docking, the crystal used by the authors was 1o8A. This is a crystal for the human testicular ACE. Why didn’t the authors use the crystal for human somatic ACE, e.g., 4APJ? The somatic ACE is much more prevalent than the testicular ACE after all?

Response: Thank you very much for your suggestion. In fact, we selected the crystal based on previous literature.  1o8A has been widely applied and used in docking experiments. (https://doi.org/10.1016/j.foodchem.2021.131070. In the future, we will conduct research based on your suggestion.   

        (vi).         Line 277: Please check whether it should be “NO” or “No”.

Response: The word “No” have been corrected to “NO”. Line 289

       (vii).         3.8 Statistical Analysis: The number of replicates is not clearly indicated. The statement “Significant level was set at 0.05” is unclear too.

Response: Statistical Analysis has been corrected to “All experiments were performed in triplicate and the results reported as the mean ± standard deviation (SD). Data were analyzed by one-way analysis of variance (ANOVA), and then Dunnett multiple tests was performed using GraphPad Prism Version 9 (San Diego, CA, USA). Significant level was set at p less than0.05.” Line 298